# NICEWEBRL: A PYTHON LIBRARY FOR HUMAN SUBJECT EXPERIMENTS WITH REINFORCEMENT LEARNING ENVIRONMENTS

## ABSTRACT

We present NiceWebRL, a research tool that enables researchers to use machine reinforcement learning (RL) environments for online human subject experiments. NiceWebRL is a Python library that allows any Jax-based environment to be transformed into an online interface, supporting both single-agent and multi-agent environments. As such, NiceWebRL enables AI researchers to compare their algorithms to human performance, cognitive scientists to test ML algorithms as theories for human cognition, and multi-agent researchers to develop algorithms for human-AI collaboration. We showcase NiceWebRL with 3 case studies that demonstrate its potential to help develop Human-like AI, Human-compatible AI, and Human-assistive AI. In the first case study (Human-like AI), NiceWebRL enables the development of a novel RL model of cognition. Here, NiceWebRL facilitates testing this model against human participants in both a grid world and Craftax, a 2D Minecraft domain. In our second case study (Human-compatible AI), NiceWebRL enables the development of a novel multi-agent RL algorithm that can generalize to human partners in the Overcooked domain. Finally, in our third case study (Human-assistive AI), we show how NiceWebRL can allow researchers to study how an LLM can assist humans on complex tasks in XLand-Minigrid, an environment with millions of hierarchical tasks. The library is available at https://anonymous.4open.science/r/nicewebrl-28BF.

## 1 INTRODUCTION

The last 20 years have seen a whirlwind of progress in Machine Learning (ML). Reinforcement Learning (RL) agents have achieved superhuman performance on complex games such as Go (Silver et al., 2017); computer vision systems can now process complex scenes (Zhou et al., 2022; Kirillov et al., 2023; Radford et al., 2021); and large language models (LLMs) increasingly act as our coding assistants and thought partners (Achiam et al., 2023; Bubeck et al., 2023).

This progress motivates many researchers to study modern Artificial Intelligence (AI) agents in the context of human behavior. Some ML researchers aim to improve AI systems by comparing them to humans, since humans can still provide an upper bound on the performance our systems can hope to achieve (Mnih et al., 2015; Team et al., 2023). For example, Minecraft (Guss et al., 2019; Hafner et al., 2023) remains a challenging exploration problem for machines but a fun exploration adventure for people (Hjorth et al., 2021; Du et al., 2023). In cognitive science, there is increased interest in asking whether these machines are human-like (Lake et al., 2017; Ying et al., 2025; Carvalho & Lampinen, 2025). Even if they are not, cognitive scientists are interested in using them as the basis for building human-like machines (Lake et al., 2017; Carvalho et al., 2025). In multi-agent RL, many researchers are interested in whether these agents can act as adaptive partners to humans across the wide range of social settings they might be deployed in (Carroll et al., 2019; Russell, 2022). This is increasingly relevant with LLMs, as they possess superhuman knowledge and are improving in their "reasoning" abilities (Guo et al., 2025). A natural question is how well they can combine their prior knowledge with environment perception to assist us in completing complex tasks. Collectively, these advances could have a wide impact—ranging from robotics (Vemprala et al., 2024) to education (Holmes & Tuomi, 2022) to healthcare (Shaheen, 2021). Clearly, many are interested in building human-like, human-compatible, and human-assistive AI.

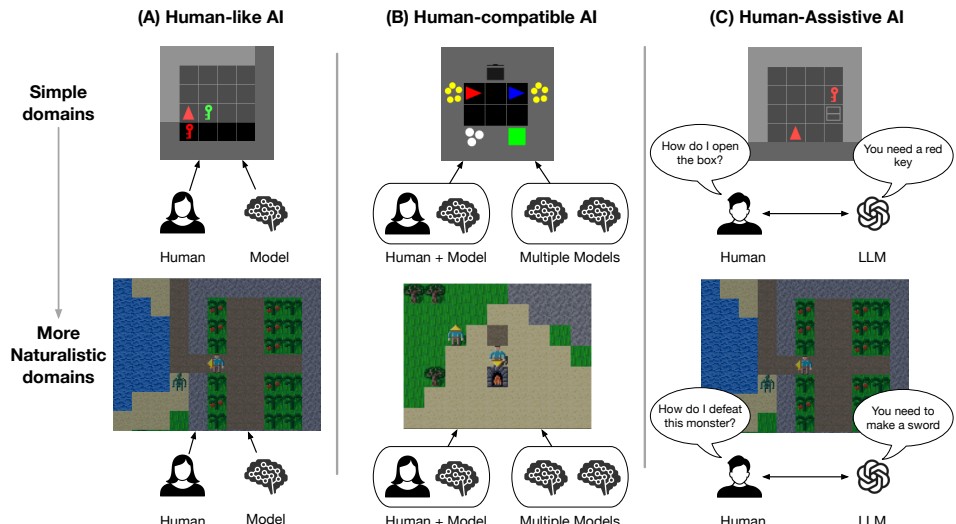

Figure 1: **NiceWebRL is a research tool for using Jax-based environments to develop Human-like, Human-compatible, and Human-assistive AI.** (A) Researchers can compare how humans and AI complete tasks to evaluate if AI behaves in human-like ways. (B) They can study how AI coordinates with humans during task completion to assess if the AI has learned human-compatible social behaviors. (C) They can also integrate Large Language Models into their experiments to evaluate how effectively they combine prior knowledge with environmental perception to assist participants. Importantly, NiceWebRL enables findings to generalize across potentially more naturalistic domains.

Despite this interest in human-centered AI development, pursuing research that integrates human subject experiments with modern ML libraries is currently a cumbersome process. To run experiments with many participants, researchers leverage the internet to get large sample sizes (Gureckis et al., 2016). Thus, most infrastructure is written in the web's programming language: JavaScript (Finger et al., 2017; Henninger et al., 2021; Gureckis et al., 2016; De Leeuw, 2015). Machine learning code, on the other hand, relies heavily on Python for model development (Abadi et al., 2016; Paszke et al., 2019; Bradbury et al., 2018) and variants of C for developing fast environments for simulation (Kolve et al., 2017; Ward et al., 2020). Leveraging ML models or environments for human subject experiments currently requires setting up domain-specific server-client configurations that integrate Javascript, Python, and sometimes C. Doing this for each domain makes the process even more cumbersome.

To address this challenge, we present NiceWebRL: a research tool that lets researchers leverage ML environments for human subject experiments (see figure 1). Integrating Python with JavaScript requires maintaining a connection between a remote Python-based server and a local Javascript-based client. This distance can cause latency issues when running online experiments. To circumvent this challenge, NiceWebRL exploits Jax (Bradbury et al., 2018)—a high-performance numerical computing library—to precompute and cache environment dynamics for arbitrary Jax-based environments. NiceWebRL then acts as a meta-environment for researchers to use arbitrary Jax-based environments in their human subject experiments. Critically, NiceWebRL allows researchers to program experiments entirely in Python by integrating with NiceGUI[1]—a library that enables web developers to specify advanced Graphical User Interface (GUI) components entirely in Python.

**Our contributions are as follows**. (1) We present NiceWebRL, a research tool that enables the use of Jax-based virtual environments for both developing artificial agents and for running human subject experiments (§4). (2) We present 3 case studies that demonstrate how NiceWebRL can support the development of Human-like AI (§5.1), Human-compatible AI (§5.2), and Human-assistive AI (§5.3). (3) Our codebase, https://anonymous.4open.science/r/nicewebrl-28BF, comes with several functional example folders using NiceWebRL across these 3 scenarios.

---

[1] https://github.com/zauberzeug/nicegui/

## 2 RELATED WORK

NiceWebRL is a meta-environment for leveraging Jax-based virtual environments in online human subject experiments. It facilitates the development of Human-like AI, Human-compatible AI, and Human-assistive AI. There is a rich literature on these topics. Researchers have created desiderata for measuring how "Human-like" AI agents are (Ying et al., 2025;?); they have developed benchmarks that test for "human-like" abilities (Zhou et al., 2023; Sclar et al., 2023); and books (Russell, 2022) and articles (Collins et al., 2024) have been written about how to enable human-compatible AI. There is a rich literature on how general AIs can assist humans (Hadfield-Menell et al., 2016; Shah et al., 2020) and a growing literature on how LLMs can assist humans on tasks (Liu et al., 2023; Wu et al., 2023). Our focus is on enabling researchers to run human subject experiments with modern ML models and environments. Below we review the most relevant literature.

**Running human subject experiments**. JavaScript is the only programming language that can be run on modern web browsers. As a consequence, most tools for human subject experiments target JavaScript-based development. For example, Labvanced (Finger et al., 2017), lab.js (Henninger et al., 2021), and Psiturk (Gureckis et al., 2016) all provide GUI interfaces for designing JavaScript-based web experiments. While there are Python-based tools for developing human subject experiments like Psychopy (Peirce, 2007), they only permit local experiments and an accompanying JavaScript library must be used for writing web experiments. JsPsych (De Leeuw, 2015) is a JavaScript library that facilitates programmatic definitions of experiments like NiceWebRL. While each are useful, none provide utility for leveraging modern ML models and environments (written in Python) for online human subject experiments. This is precisely the gap that NiceWebRL aims to fill.

**Comparing natural intelligence and artificial intelligence in virtual environments**. Many of the efforts that compare natural and artificial intelligence rely on the Unity game engine, which allows for the development of 3D environments with realistic physics and sensory observations (Ward et al., 2020). Cobel-RL developed a 3D environment for studying neuroscience questions around spatial navigation with Deep learning based RL models (i.e. Deep RL models) (Diekmann et al., 2023). The Animal-AI environment was developed to study whether Deep RL models displayed cognitive abilities associated with animals (Beyret et al., 2019). Most similar to NiceWebRL is PsychLab which explicitly aims to compare Humans to Deep RL models across several classic experimental paradigms like visual search, multiple object tracking, and random dot motion discrimination (Leibo et al., 2018). However, the following challenges its adoption. (1) Its API is in Lua, which limits accessibility, whereas NiceWebRL is purely in Python, (2) it lacks precise response time measurements which NiceWebRL provides, (3) its reliance on Unity makes it challenging to run online experiments. While NiceWebRL relies on Python, it is able to run online but maintain fast environment performance thanks to its reliance on Jax.

## 3 BACKGROUND

**The Jax ecosystem**. Jax (Bradbury et al., 2018) is a Python library for high-performance numerical computing and machine learning with NumPy-like syntax. It follows a functional programming paradigm where functions are stateless, only defining input transformations. It achieves fast computation through JIT compilation and includes tools for tracking random number generators, enhancing reproducibility. Most relevant to NiceWebRL, there is a growing set of environments including XLand-Minigrid, which has millions of hierarchical tasks for studying long-horizon generalization (Nikulin et al., 2024); JaxMARL, which has multi-agent environments for studying coordination (Rutherford et al., 2024); Pgx, which has planning environments for studying reasoning (Koyamada et al., 2023); Craftax, a large procedurally-generated open-world environment that enables studying

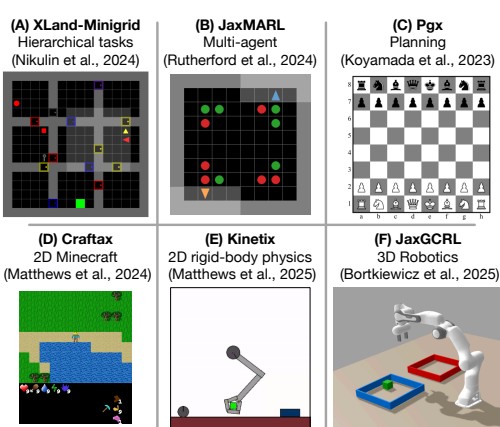

Figure 2: **Examples of Jax-based environments in the Jax ecosystem.** All of these can be leveraged with NiceWebRL to develop Human-like, Human-compatible, or Human-assistive AI.

exploration and generalization (Matthews et al., 2024); Kinetix, a procedurally-generated physics-based environment for studying physical reasoning (Matthews et al., 2025); and JaxGCRL, a goal-conditioned robotics environment for studying 3D manipulation tasks (Bortkiewicz et al., 2025). NiceWebRL can be used with all of these environments.

**Python-based web development** requires maintaining a Python-based web server that communicates with clients that operate in JavaScript. We build on NiceGUI which comes with tools for handling many concurrent client connections asynchronously. When sending large data packets between a client and a server, web socket connections are needed for real-time communication. NiceGUI's web socket implementation facilitates setting up persistent connections by having web sockets automatically reopen when connections close unexpectedly. This is key to having seamless human experiments with Python-based environment backends. Finally, and equally important for online experiments, NiceGUI enables researchers to use Python to build responsive GUI components without any JavaScript knowledge. We provide examples of GUI components provided by NiceGUI in figure 3. All of these can be used with NiceWebRL.

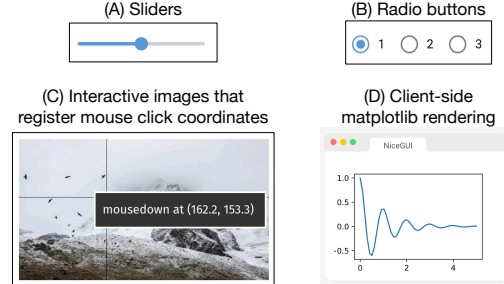

Figure 3: **Examples of advanced GUI capabilities provided by NiceGUI that can be leveraged with NiceWebRL.** Researchers can create (A) sliders for reporting numerical scores (B) radio buttons for selecting choices. (C) Beyond recording key strokes, researchers can record actions as $(x, y)$-coordinate selections on images or (D) leverage familiar plotting tools such as matplotlib for presenting graphs to participants.

## 4 NICEWEBRL

NiceWebRL is a Python library that leverages Jax and NiceGUI to create online interfaces for human interaction with Jax-based virtual environments. NiceWebRL allows for any Jax-based environment as input and allows participants to interact with the environment until some criteria is met (e.g. a minimum number of successful episodes). To obtain different environment behaviors across different parts of an experiment, NiceWebRL leverages "environment parameters" (e.g. a fixed task distribution or friction coefficient) to define different dynamics based on a user's experiment stage. NiceWebRL leverages a functional paradigm to decouple (1) the function that defines how the environment will reset and evolve from (2) information about the participant's state and from the environment parameters. This design allows a single compiled environment to be used across different experiment stages by different participants with their own independent environment state. We present an overview in figure 4. We describe our setup more formally in §A and provide more details on and examples of stage types in §B.

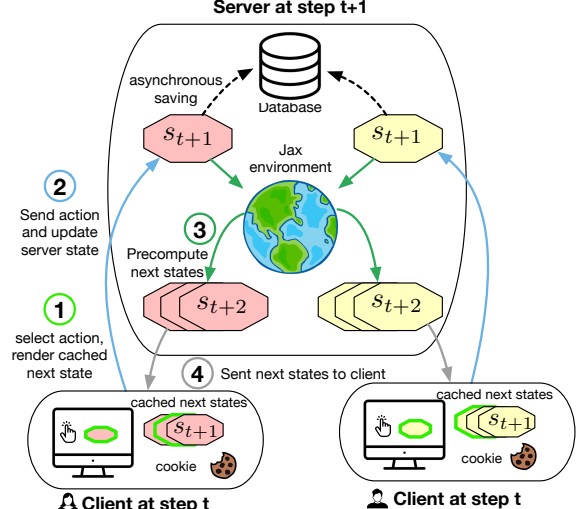

Figure 4: **How NiceWebRL leverages Jax to enable a single server to interact with multiple users**. Jax's functional paradigm prohibits inter-user state interference since each user has isolated environment states. Jax enables fast parallel computation of future states. Caching these states client side reduces latency.

**Reducing latency when presenting observations generated by a remote server.** NiceWebRL precomputes potential next states to reduce latency. When a stage initializes interaction with a participant ($t = 0$), the server computes the initial state $s_0$ and observation $o_0$, then immediately computes all possible next states $\{s_1\}$ and observations $\{o_1\}$ for each potential action $a$. Both the initial observation and all potential next observations are

sent to the client before any user interaction occurs. When the observation renders, time $t_1$ is recorded. When the participant selects an action $a_0$, time $t_2$ is recorded. The client immediately renders the corresponding precomputed observation $o_1$ and sends $(a_0, t_1, t_2)$ to the server. The server then selects the appropriate state from its cache, computes the next set of potential states and observations for all possible actions, and sends these to the client for the next interaction. This parallel computation of all potential next states and observations, enabled by JAX, reduces network latency and helps provide immediate visual feedback to participants. We summarize this server-client Human-environment interaction protocol in Algorithm 9.

**Data management and persistence.** Having a persistent participant state across page reloads or web socket connection issues is key to a fluid experiment. We maintain this in two ways. First, we leverage NiceGUI to track unique identifier information held in browser session cookies. To track a participant's experiment progress, we exploit Jax's functional paradigm to track a user's environment state and current random number generator. We automatically serialize these objects and store them for every environment-interaction in a SQL database. Whenever a connection is reset, we identify a user and reload their information for fluid re-engagement with the experiment. When a researcher wants to later analyze the experiment data, they have access to all interaction data in its native format.

**Reducing latency when serving multiple clients**. When serving multiple clients, performing I/O operations for persistence can be resource intensive and block other operations. To mitigate this, we save data asynchronously with a queue-based strategy that leverages stochastic exponential backoffs whenever a save fails. That is, upon database integrity errors, the system retries with delays following $d_i = \min(\text{basedelay} \times 2^i + \mathcal{U}(0, 0.1), \text{maxdelay})$ where $i$ is the attempt number, and $\mathcal{U}(0, 0.1)$ adds random jitter to prevent synchronized collisions if multiple participants try to save concurrently. This is important for having a responsive UI when the server experiences many parallel participants.

**Human-AI coordination.** When a human coordinates with an artificial agent, we can have the agent be a part of the environment. When the environment steps, it not only computes the state and observation, it also computes the action that the artificial agents would compute for that state and uses that to predict all possible next states for the participant. Thanks to Jax, environments and learning-based agents can be compiled into one function, reducing latency from this extra computation.

**AI-assisted task completion.** This is a single-agent setting where an LLM has access to either the environment state $s_t$ or environment observation $o_t$. Two options for leveraging LLMs exist. One option is to interact with the LLM via API calls. A second option is to use a local LLM. We provide examples of each in our examples folder.

## 5 CASE STUDIES

We present three case studies that display how NiceWebRL can help in the development of human-like, human-compatible, and human-assistive AI. In §5.1 and §5.2, we'll highlight how NiceWebRL contributed to new insights that span AI and cognitive science research on human-like and human-compatible AI models. We will first describe the experimental results of prior work and then discuss how this is made possible by NiceWebRL. In §5.3, we will present a proof-of-concept for how NiceWebRL can support research on LLM-based human-assistive AI. We emphasize that all case studies have accompanying code in the repository.

### 5.1 CASE STUDY 1: DEVELOPING HUMAN-LIKE AI WITH NICEWEBRL

**Developing a novel Deep RL cognitive science model with NiceWebRL (Carvalho et al., 2025).** A central question in cognitive science is how people represent the environment to enable generalization to new tasks. Successor features (SFs) are a mechanism for how an agent can cache expectations of what it will see when pursuing a policy (Barreto et al., 2017). Recent ML research has shown that SFs enable agents to repurpose policies for new tasks (Barreto et al., 2018; Carvalho et al., 2024). Later, cognitive scientists showed that SFs also explained how people reuse prior policies for new tasks (Tomov et al., 2021). However, the behavioral work was done in a small grid-world with 13 states. Carvalho et al. (2025) studied whether SFs could explain how humans reuse behaviors for new tasks in 2 more complex domains: a maze gridworld (figure 5 A) and Craftax (Matthews et al., 2024), a 2D minecraft domain (figure 5 D). Across both domains, they set up training tasks where a test object was visible from along the optimal training paths (e.g. top-right corner of figure 5 A). SFs could not generalize here. People could; however, when people reused a training path to

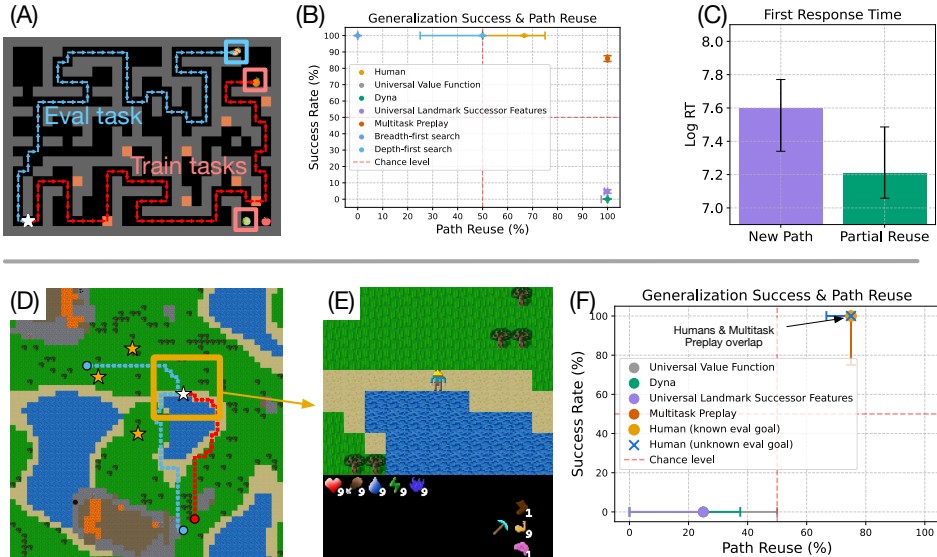

Figure 5: **Case study 1: NiceWebRL enabled the development of a novel Deep RL cognitive model that generalizes to new tasks with the same qualitative behaviors as Humans across multiple domains**. (A, D): a gridworld and 2D minecraft environment that both Human participants and Deep RL models learned in. (B, F): behavioral results studying the same phenomena across the two domains. NiceWebRL enabled developing a cognitive model that (1) generalized to novel test goals in ways not permitted by previous methods, while (2) doing so with a similar suboptimal path reuse strategy that humans tend to exhibit. Figures reproduced from (Carvalho et al., 2025).

a novel goal, their response times suggested that they were using a caching-based solution rather than something more flexible—but expensive—like planning at decision-time (figure 5 B-C). They developed an algorithm termed Multitask Preplay which *preemptively* learns solutions for unpursued tasks nearby training tasks by augmenting experience replay with small amounts of counterfactual simulation. They found this algorithm both better accounted for the response times people exhibited and better predicted how they would reuse prior behaviors. These results generalized to Craftax, where participants and models had to navigate from partial observations of a large world with many objects (figure 5 E-F).

**Role of NiceWebRL.** First, NiceWebRL enabled comparing human behavior to advanced Deep RL algorithms including Successor Features. Second, NiceWebRL enabled using the same infrastructure to study both Deep RL algorithms and human behavior in two domains of increasing complexity: a gridworld and Craftax. This helped to ensure that findings were generalizable. Finally, NiceWebRL enabled the measurement of response times. This helped to adjudicate between theories that predict more or less computation at decision-time.

### 5.2 CASE STUDY 2: DEVELOPING HUMAN-COMPATIBLE AI WITH NICEWEBRL

**Developing a novel Multi-agent reinforcement learning (MARL) algorithm for coordinating with humans using NiceWebRL (Jha et al., 2025)**. One central question in MARL is how we can develop MARL agents that can generalize to human partners without human training data. One current benchmark for human-compatible AI is the Overcooked domain (Carroll et al., 2019) where agents must coordinate on basic cooking tasks. The state-of-the art algorithm is "Efficient End-to-End Training" (E3T; Yan et al., 2023), a "Self Play" algorithm that plays with—and tries to predict the actions of—a noisy variant of itself. Jha et al. (2025) developed a novel algorithm, Cross-Environment Cooperation (CEC), where an agent plays only against itself but across millions of different procedurally-generated environments (figure 6 A). They found that while E3T (Yan et al., 2023) was able to "succeed" on more episodes when collaborating with humans (figure 6 B), humans gave that agent a lower rating than CEC (figure 6 C). The authors asked participants questions about their subjective experience using a Likert scale (Likert, 1932) and found that CEC was rated as more "adaptive" and "human-like"—despite succeeding *less* than E3T. When the authors analyzed game trajectories, they found CEC would collide less with humans across environments.

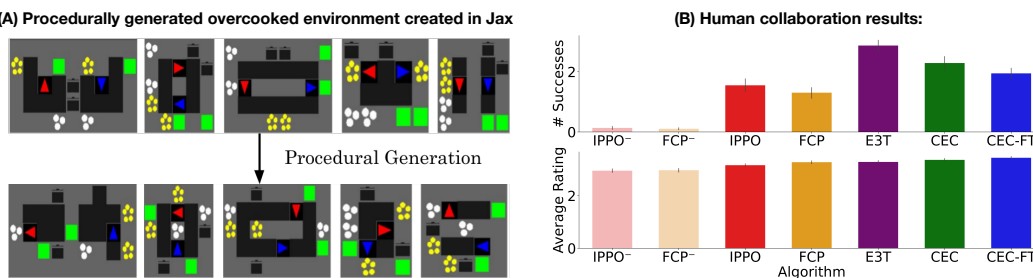

Figure 6: **Case study 2: NiceWebRL enabled the development of a novel MARL algorithm that is more compatible with novel human partners.** (A) A procedurally-generated environment used to design a novel MARL algorithm: Cross-Environment Cooperation (CEC). Prior work had agents learn with diverse sets of agents. CEC has a *single agent* play itself across millions of procedurally-generated environments. (B) While CEC *succeeded less* than other methods when collaborating with humans (top), it succeeded in ways that were *most favorable* to humans (bottom). Analysis suggested showed prior agents succeeded in less collaborative ways. Figures reproduced from (Jha et al., 2025).

**Role of NiceWebRL.** First, NiceWebRL enabled comparing multiple MARL algorithms in their ability to generalize to human partners. Second, NiceWebRL enabled the researchers to collect feedback from participants after every environment interaction stage using the "Feedback" stage object available in NiceWebRL coupled with NiceGUI's data collection GUI elements. This provided an easy way to get feedback from participants while agent-interaction data was fresh in their memories. Third, NiceWebRL stores all environment interactions so participant episodes could be analyzed post-hoc. This enabled the researchers to analyze trajectories by participants and agents to determine what qualitative behaviors (such as colliding) were different between the different MARL algorithms.

### 5.3 CASE STUDY 3: DEVELOPING HUMAN-ASSISTIVE AI WITH NICEWEBRL

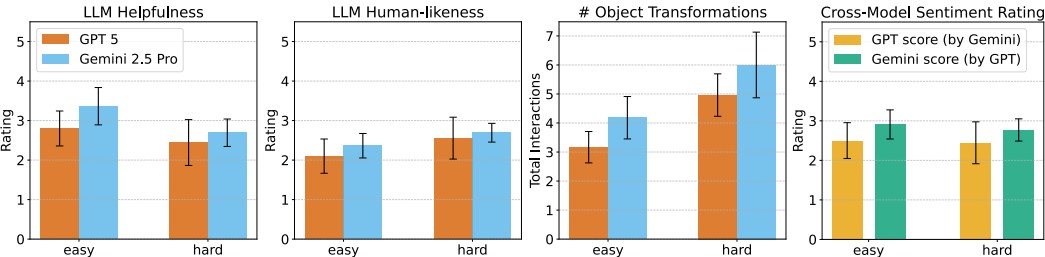

Figure 7: **Case study 3: Proof-of-concept experiment showing that NiceWebRL enables comparing how different LLMs can assist people in completing tasks.** We had GPT 5 or Gemini 2.5 Pro act as assistants for people completing tasks in the XLand-Minigrid domain (Nikulin et al., 2024). Each plot is showing the mean and standard error for 10 subjects per model.

**Developing an LLM-assistant for sequential-decision making tasks in a virtual environment**. We created a simple proof-of-concept experiment where people had to interact with "easy" and "hard" tasks from the Xland-Minigrid domain (Nikulin et al., 2024). To require assistance, they were given no task information but could ask an (anonymous) LLM assistant for help. At the beginning of the experiment, users were randomly assigned either GPT 5 or Gemini 2.5 Pro. We set up our server so that it would interact with the LLMs via API calls[2]. Importantly, the LLM assistants were given text descriptions of the ground truth environment state including information on: (1) the goal of the episode (2) the locations and identities of all objects in the environment (3) the rules of the environment (e.g. how objects interact when combined). In principle, this can enable them to help users figure out the goal to maximize task reward. In this proof-of-concept, each participant completed 3 episodes, where a new task was sampled per episode. After all 3 episodes, participants answered two questions on a 5-point scale, "How helpful was the AI?" and "How human-like was the AI?". We collect data from 40 participants via CloudResearch. We describe details around recruiting participants in §E. We show results in figure 7.

---

[2]we provide an example of how to set up a web experiment with a local LLM in our examples folder

**Role of NiceWebRL**. NiceWebRL enabled using an existing ML domain to develop an experiment that studied how an LLM could assist people on long-horizon tasks. Additionally, the Feedback Stage object in NiceWebRL enabled collecting feedback from participants about the LLMs at the end of the experiment. This could also be done after every episode or during episodes. We leveraged this feedback in post analysis to predict the sentiment of users by the "opposing" LLM.

## 6 SPEED AND MEMORY PERFORMANCE

**Speed performance**. NiceWebRL's speed benefits come from the key strategies NiceWebRL uses to improve latency. In summary, NiceWebRL uses **JIT** to compile machine learning environment computations into machine code, it uses **VMAP** for parallel precomputation of next states, and it preemptively **caches** next state information client side. To understand the benefits of these, we ablate them. Notice that VMAP *supports* caching. To ablate this, we consider both the condition where there is no caching at all, and the condition where there is caching but it is supported by parallel computation via multiprocessing.

To understand the kind of latency users experience, we consider two measurements. The first is "Image seen time", the time between when a key is pressed and when the next image is seen, i.e. $\Delta$ (Key Press, Image Seen). This is *trivially* 0 for all caching based solutions. The next measurement, "System readiness", is the time from when a key is pressed client side to when the system has all information from the server to accept the next key (including travel time between the server and client), i.e. $\Delta$ (Key press, Accepts keys again). If a system is fast, then this number should be mainly bottlenecked by server-client latency. To obtain these measurements, we setup a remote server and then had a client interact with the server at regular intervals. For each measurement, we collected 50 interactions.

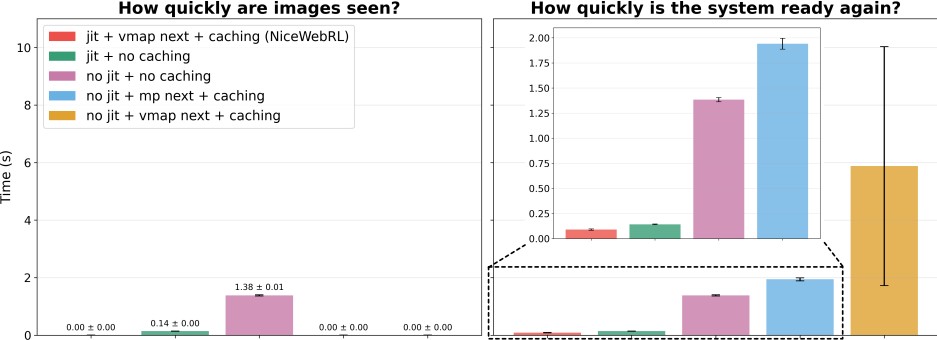

Figure 8: **Combining JIT, VMAP, and CACHING leads to the fastest system response time.** We see that JIT by itself dramatically lowers system readiness time to $< .2$ seconds. Without JIT, we see that preemptive caching is not tenable at over 1 second. Interestingly, parallel computation via multiprocessing is *slower* than not precomputing at all (we discuss this in §7.1).

**Memory performance**. We also look at how NiceWebRL memory performance scales with the number of users. We measure Resident Set Size (RSS), which represents the portion of a process's memory that is held in physical RAM and excludes swapped or shared memory pages. For JAX-based applications like NiceWebRL, RSS growth typically reflects the memory cost of maintaining separate environment states and cached computations for each concurrent user. We find that, unsurprisingly, memory usage scales linearly with users (see figure 12). This means that we are memory bounded. One can circumvent this by spreading demand across multiple servers. With the advent of ML in industry, many services offer this service

## 7 DISCUSSION AND CONCLUSION

We have presented NiceWebRL, a library for writing human subject experiments that leverage machine learning models and environments. Importantly, by integrating with NiceGUI, experiments with sophisticated GUI components can be written entirely in Python. NiceWebRL exploits Jax's compilation features and functional programming paradigm to reduce latency and enable multiple clients to interact with a single backend server. We demonstrated the utility of NiceWebRL with three case

studies spanning both single-agent and multi-agent settings across 4 domains: a custom gridworld, Craftax (Matthews et al., 2024), Overcooked (Rutherford et al., 2024), and Xland-Minigrid (Nikulin et al., 2024). In the first case study, we showed that NiceWebRL could be used to measure human task performance and response times when developing a cognitive science model that could predict human behavior across two domains. In the second case study, we showed that NiceWebRL could be used to compare different MARL algorithms in their ability to generalize to humans without human training data. Here, we showed that NiceWebRL's stage objects facilitated qualitative and quantitative analysis studying why different algorithms were rated to be better collaborators by human participants. In our final case study, we showed that one could also use NiceWebRL to run experiments that measure how well LLMs can assist people on sequential decision-making tasks under asymmetric information constraints.

## 7.1 The importance of Jax

Our results in §6 highlight the importance of leveraging Jax for python-based backend servers hosting human subject experiments with machine learning environments. Without Jax, multiprocessing introduces significant complexity in multi-user web applications: each user session operates in its own thread, and when multiple users simultaneously trigger parallel computations, they spawn separate process pools. This creates resource contention as processes compete for CPU cores, memory overhead from duplicating environment information across processes, serialization bottlenecks from pickling/unpickling environment objects, and potential deadlocks when multiple process pools acquire system resources simultaneously. JAX sidesteps these problems by using compiler-level vectorization to compute multiple environment steps in parallel on the same accelerator within a single process.

## 7.2 What else is possible with NiceWebRL?

**Scaling up human-AI comparisons**. By leveraging the *same* environment for both human and AI interaction, this facilitates developing standardized data formats across human and AI data. Standardized data formats facilitate testing the *generalizability* of a learning or decision-making algorithm. That is, not just does my algorithm learn behavior that generalizes to new data, but does my method for learning or decision-making *itself* generalize to new domains. Generalizability-focused has been key in developing general ML methods. It may likewise be key in developing general human-centric AI methods as well.

**Event-triggered experiment progression**. NiceWebRL allows environment events to control experiment flow. First, events can trigger data collection (forms, rating scales, text boxes, etc.) directly in Python via NiceGUI. Since NiceWebRL uses Jax, triggers can also include complex neural network functions. For example, one could use diffusion-model-supported monte carlo tree search (Yoon et al., 2025) to exploit fast predictions of what participants *might do* to define triggers.

## 7.3 Limitations

While we've demonstrated NiceWebRL's utility for human subject experiments, many improvement avenues remain. We don't currently leverage Jax-based environment's ability to allow gradients to pass through their computational graph. Unsupervised environment design (Dennis et al., 2020) can exploit this to automatically generate environments with different properties. Like vision researchers use gradient descent to generate stimuli for humans (Geirhos et al., 2018) and monkeys (Wang & Ponce, 2022), NiceWebRL may be able to automatically generate environments for different target experimental conditions. Another limitation is that NiceWebRL currently only supports multi-agent domains with 2 agents. Future work can look to generalize our library to the $n$-dimensional case. Finally, while we precompute all next states to reduce latency, this may become prohibitive for large or continuous action spaces. Future work can integrate policy learning within the environment to select likely human actions for precomputing next states. Jax enables this policy to be incorporated into the environment's computational graph, which minimizes its computational cost.

NiceWebRL's reliance on Jax allows for a rich set of tools to improve future experiments. Jax has a growing ecosystem of libraries spanning probabilistic programming Bingham et al. (2019), Bayesian inference (Cabezas et al., 2024), LLM development (Geng, 2023), and general neural network development (Heek et al., 2024; Kidger & Garcia, 2021). This enables researchers from various disciplines to leverage NiceWebRL for human subject experiments regardless of their preferred modeling approach. Thus, we are optimistic that NiceWebRL can serve as a useful tool for future research developing Human-like, Human-compatible, and Human-assistive AI.

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
