# A   FORMAL DOMAIN DESCRIPTION

Jax functions automatically compile to a fixed behavior when they receive their first input data. As such, if one wants different domain functionality across different *contexts* (e.g. training vs. testing), the domain's functions typically need a "env_parameter" argument. Thus, Jax-based domains are naturally formulated as Partially Observable Contextual Markov Decision Processes (POCMDPs) $\mathcal{M}_c = \langle S, \mathcal{A}, \mathcal{X}, \mathcal{C}, \rho, P, R, O \rangle$ (Hallak et al., 2015; Kaelbling et al., 1998). Here, $S$ denotes the environment state space, $\mathcal{A}$ denotes its action space, $\mathcal{X}$ denotes (potentially partial) observations of the environment, and $\mathcal{C}$ denotes a space of contexts that an MDP can be in. env_parameter then corresponds to an MDP's context $c \in \mathcal{C}$. It can be used to augment the initial state distribution $\rho_c(s_0)$ (e.g. having an agent start in different states in different contexts), the transition probabilities, $P_c(s'|s, a)$ (e.g. an agent's speed or strength can be changed in different contexts), the reward function $R_c(s)$ (e.g. different objects can be rewarded in different contexts), or the observation function $O_c(s)$ (e.g. objects can take on different colors in different contexts).

An episode proceeds as follows. An initial state $s_0 \in S$ is sampled from the initial state distribution $\rho_c(s_0)$. When an agent takes an action $a \in \mathcal{A}$ in state $s \in S$, the next state $s'$ is sampled according to a next state distribution $s' \sim P_c(\cdot|s, a)$. The agent then receives an observation $x' = O_c(s')$ and reward $r' = R_c(o)$. Note that $c$ is typically fixed within an episode.

| **Server-side Operations** | **Client-side Operations** |
|---|---|
| Input: env context parameters $c$ | |
| | |
| At time $t = 0$ : | |
| 1: $s_0, o_0 = $ env.reset($c$) | 1: |
| 2: $\{(s_1, o_1) = $ env.step($s_0, a, c$)$\}_{a \in \mathcal{A}}$ | 2: |
| 3: cache $s_{\text{next}} = \{s_1\}$ | 3: |
| 4: send $o_0$ and $o_{\text{next}} = \{o_1\}$ to the client | 4: display $o_0$ and record time $t_1$ |
| 5: | 5: cache $o_{\text{next}}$ |
| 6: | 6: participant selects action $a$ |
| 7: | 7: record time $t_2$ |
| 8: | 8: send $a_0, t_1, t_2$ to the server |
| 9: | 9: select $o_1 \in o_{\text{next}}$ corresponding to $a_0$ |
| 10: | 10: display $o_1$ and record $t_1$ |
| | |
| At time $t = 1, 2, \dots$ : | |
| 11: receive and store $(a_{t-1}, t_1, t_2)$ | 11: |
| 12: select $s_t \in s_{\text{next}}$ corresponding to $a_t$ | 12: |
| 13: $\{s_{t+1}, o_{t+1} = $ env.step($s_t, a, c$)$\}_{a \in \mathcal{A}}$ | 13: |
| 14: send $o_{\text{next}} = \{o_{t+1}\}$ to the client | 14: cache $o_{\text{next}}$ |
| 15: update $s_{\text{next}} = \{s_{t+1}\}$ | 15: participant selects $a_t$ |
| | 16: record time $t_2$ |
| | 17: send $a_t, t_1, t_2$ to the server |
| | 18: select $o_{t+1} \in o_{\text{next}}$ corresponding to $a_t$ |
| | 19: display $o_{t+1}$ and record $t_1$ |

Figure 9: **Server-client Human-Environment Interaction Protocol.** Note that we omit displaying reward $r$ due to space constraints.

Let env be the programmatic object representing a domain. In our library (and many RL libraries), $s_0, o_0 = $ env.step($c$) essentially plays the role of sampling from the initial state distribution and computing the corresponding observation for the agent. The standard practice is to have $s_{t+1}, o_{t+1}, r_{t+1} = $ env.step($s_t, a_t, c$) implement (a) sampling a new state (b) computing the corresponding reward, (c) computing the observation that an agent will get.

# B   DESCRIPTIONS OF STAGE TYPES

Currently, there are three basic stage classes, though more can easily be added.

1. `Stage`: used to display instructions or information to a participant.

2. `FeedbackStage`: used to collect information from participants. Typically involves an interactive screen that does *not* interact with the environment.

3. `EnvStage`: used to interact with an environment. It takes as input an environment and environment parameters. We describe how NiceWebRL uses this abstraction to have a remote server-side program display images to one's local web-browser client in figure 9.

We present examples of each in figure 10.

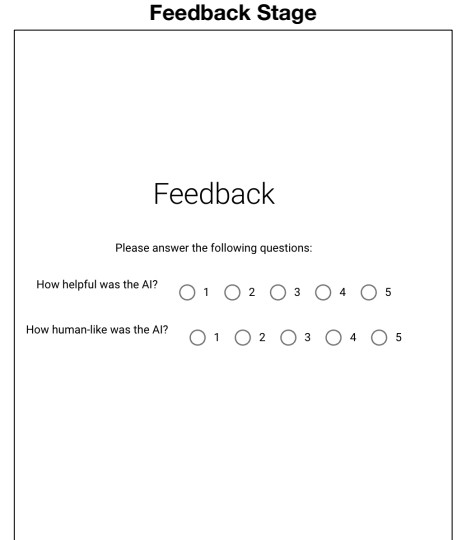

Figure 10: **Examples of different kinds of stages.**

## C    ADDITIONAL RESULTS

## D    COMPUTING RESOURCES

For details on case study 1 or 2, please see (Carvalho et al., 2025) or (Jha et al., 2025), respectively. For case study 3, experiments were conducted using computing infrastructure from the fly.io platform with the "performance-2x" configuration. This is a machine with 4GB of RAM. The

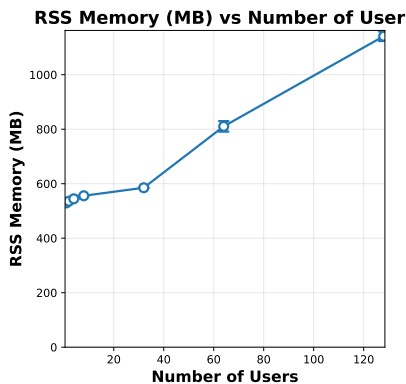

Figure 11: **Memory usage vs. number of users**

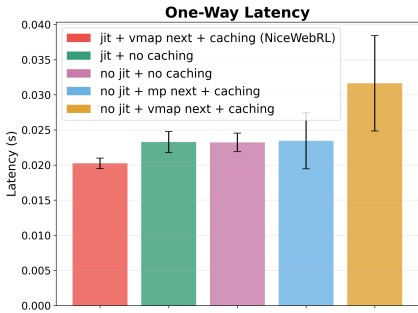

Figure 12: **Round Trip Latency**

machine had no GPU. Even in this setting, Jax's compilation features provide a significant speed up to environment computation.

## E  HUMAN SUBJECT EXPERIMENT DETAILS

Our study is approved by the University IRB. All subjects were recruited with https://www.cloudresearch.com/ and provided informed consent. We provide the consent form in the GitHub example. Participants were compensated $4 for completing the task. The average task completion time was 23.33 minutes. At the beginning of each experiment, the participants provided demographic information (age and gender, coded as male or female).