# OpenReview forum: "NiceWebRL: a Python library for human subject experiments with reinforcement learning environments"
_ICLR.cc/2026/Conference — ICLR 2026 Conference Withdrawn Submission_

### Official Review · Reviewer_QRRe · 2025-10-31

**Soundness:** 2
**Presentation:** 3
**Contribution:** 2
**Rating:** 4
**Confidence:** 4

**Summary:**

The paper introduces NiceWebRL, a Python library that enables researchers to conduct online human subject experiments using JAX-based reinforcement learning environments. The tool supports single-agent scenarios for comparing human and AI performance, multi-agent settings for studying cooperation, and human-AI assistance paradigms. The authors leverage JAX's functional programming paradigm and JIT compilation to precompute and cache environment dynamics, enabling low-latency interactions through a centralized server architecture. Three case studies demonstrate applications in developing human-like AI (cognitive modeling), human-compatible AI (multi-agent coordination), and human-assistive AI (LLM-based assistance).

**Strengths:**

- Comprehensive literature review: The authors provide valuable context on existing tools for human subject experiments, clearly identifying the gap between JavaScript-based experimental frameworks and Python-based ML environments.
- Clear conceptual framework (Figure 1): Excellent visual overview of different paradigms for integrating humans into RL environments, spanning human-like, human-compatible, and human-assistive AI research.
- Timely integration of LLMs: The incorporation of LLM assistants into experimental paradigms (Case Study 3) addresses current research interests in human-AI collaboration with language models.
- Well-executed case studies: Three concrete applications with published results (Carvalho et al., 2025; Jha et al., 2025) demonstrate real research impact, showing how NiceWebRL enables novel cognitive science and MARL findings.
- Practical implementation details: Thoughtful features like automatic state persistence, asynchronous database operations, and browser session management show attention to real experimental needs.
- Open-source contribution: The library comes with functional examples and clear documentation, lowering the barrier for adoption.

**Weaknesses:**

- Questionable performance justification: The authors repeatedly emphasize JAX's speed advantages (lines 91, 221, 242), but fail to justify why this matters for human experiments. JavaScript and Unity/C# environments already provide sub-millisecond response times, which far exceed human reaction capabilities (typically 200-300ms).
- Artificial JAX limitation: The restriction to JAX-based environments appears unmotivated. The core contribution -- enabling human-subject experiments -- doesn't inherently require JAX. This limitation excludes popular environments like Unity-based simulations without clear justification.
- Self-imposed architectural bottleneck: The centralized server design (Section 6) creates the very performance problem that JAX then solves. A distributed architecture with client-side environment computation would eliminate this bottleneck entirely while maintaining data collection capabilities.
- Engineering overshadows science: Substantial portions of the paper focus on JAX-specific implementation details rather than the scientific challenges of human-AI interaction research. The emphasis on parallel computation and JIT compilation feels misplaced for a human-subjects tool.
- Narrow scope of generalizability: While the authors claim to address "modern ML environments," they actually only support a subset of JAX-based environments, limiting applicability to the broader research community.
- Incomplete performance analysis: The ablation study (Figure 8) doesn't compare against established alternatives like local JavaScript implementations or Unity-based systems, making it difficult to assess the actual performance benefits.
- Missing alternative solutions: No discussion of existing high-throughput environment libraries (e.g., PufferLib) that achieve similar performance without JAX dependency.

**Questions:**

- Line 113: Missing citation shown as "?" - please provide the appropriate reference.
- Figure 4: Is "Database" incorrectly cropped, or is this a rendering issue?
- Line 261: Why specifically use successor features in the cognitive modeling example? This choice seems arbitrary and unexplained.
- Performance requirements: What empirical evidence supports the need for sub-200ms response times in human subject experiments? Human reaction times typically exceed this by an order of magnitude.
- Architecture justification: Why choose a centralized server architecture that processes all computations for multiple users (e.g., 50 concurrent users) rather than client-side computation with centralized data collection?
- JAX alternatives: Have you considered supporting other high-performance environment frameworks like PufferLib that don't require JAX?
- Scalability limits: What happens when the number of concurrent users exceeds server capacity? How does the system degrade?
- Environment conversion: What effort is required to convert an existing non-JAX environment to work with NiceWebRL?

---

### Official Review · Reviewer_ULhC · 2025-10-31

**Soundness:** 2
**Presentation:** 3
**Contribution:** 2
**Rating:** 4
**Confidence:** 3

**Summary:**

The paper introduces NiceWebRL, a Python library that integrates JAX-based RL environments with NiceGUI to run online human-subject experiments with low latency via JIT compilation, VMAP precomputation and client-side caching. It provides a meta-environment abstraction and stage-based experiment design to support single- and multi-agent studies, human–AI coordination and LLM-assisted settings. The authors use three case studies demonstrate utility.

**Strengths:**

1. It is a practical tool/library that unifies JAX envs and a web GUI.
2. The experimental validation around latency and memory scaling is good. Architecture often grounded in JAX.
3. Has useful widgets and feedback built-in.
4. Code is provided.

**Weaknesses:**

1. The case studies rely on fairly small samples, limited trial counts and lack reporting of statistical significance tests (e.g. p-values, effect sizes) which limits the strength of the authors' claims.
2. As this is a library / tool, it should be compared to the most similar tools. I would have expected head-to-head system-level benchmarking vs. alternative platforms (e.g., PsychLab, jsPsych+Python bridges) for latency, throughput or developer effort.
3. It is only focused on JAX-based environments and portability to PyTorch/Unity is discussed qualitatively but not really validated.
4. Being a python library paper makes it harder to judge against typical ICLR criteria. But I would have liked to have seen either direct quantitative comparisons with related work or more quantitative experiments in the case studies.
5. I am unsure if this falls within the CFP for ICLR. I see arguments both for (supports ML/RL research) and against it (software package, and the case studies lack some research attributes like proper statistical analysis of the tool, see above) but I am open to hearing more arguments on this; and, hopefully, other reviewers will pitch in here.

### Minor
- L113, missing citation
- “Figure” in text doesn’t have consistent capitalisation
- Inconsistent Jax/JAX usage

**Questions:**

1. For each case study, what are per-condition Ns, trial/episode counts per participant, exclusion criteria, and participant demographics? Any preregistration?
2. In overcooked, how were E3T and other baselines implemented/tuned (hyperparameters, seeds, training compute)? Were environments/layouts identical? How were order effects and learning effects controlled for human partners?
3. Which browsers are supported? Mobile/desktop? Screen-reader/accessibility support?
4. What were the consent procedures, compensation, data retention and privacy safeguards for storing trajectories and chat logs. IRB approval status for each study?
5. I noticed in one part it says “40 participants” and in another it claims the participants were “10 per model”, were there 4 models?

---

### Official Review · Reviewer_bWwj · 2025-10-31

**Soundness:** 3
**Presentation:** 4
**Contribution:** 2
**Rating:** 6
**Confidence:** 3

**Summary:**

This paper introduces NiceWebRL, a library for transforming JAX environments into a web interface, this allow researchers to compare AI performance to human participants as well as human AI collaboration. The paper provides three case studies where NiceWebRL has been used to forward scientific research in human AI interaction.

**Strengths:**

This paper is very well written and seems like a useful tool for researchers. The proposed library would be of value to the research community. The technical implementation seem thoughtfully engineered and robust, making use of JAX's unique characteristics for efficiency and lowering latency. This seems to fill a practical gap in the AI and cognitive science ecosystems. The presented case studies enabled by the tool do seem of value to the scientific community.

**Weaknesses:**

Whilst a useful and well engineered tool for research, this work doesn't contribute any new test settings, acting mostly as an interface for existing environments. Therefore the tool itself seems incremental. I think expanding the comparison between LLM agents in 5.3 (case study 3) beyond just gpt-5 and Gemini 2.5 pro would be of interest, although the authors acknowledge this as a proof of concept. Given the memory boundedness as as the number of users increases, the paper could also benefit from experiments under scale to examine a little further into this weakness. For instance, detailing the impact of hardware (since it seems to also work well without a GPU), or scaling action spaces/environment complexity. Whilst capitalizing on the performance benefits of JAX brings computational advantages, it limits applicability to the majority of environments which use other frameworks, therefore additional support for PyTorch would increase potential use cases. Since 7.1 attributes performance to JAX's advantages in computation, there may be some warrant for comparison. I.e. Why not use torch.compile, torch.vmap etc

**Questions:**

1. Is there any additional advantage when using a JAX based local LLM or another LLM which runs on GPU? I.e. are you able to jit inference and the environment together?
2. Are there any data privacy issues or ethical considerations for human subjects?

---

### Official Review · Reviewer_hBa6 · 2025-10-31

**Soundness:** 4
**Presentation:** 4
**Contribution:** 3
**Rating:** 6
**Confidence:** 3

**Summary:**

This paper presents NiceWebRL, a python library that allows researchers to easily deploy Jax-based RL environments for online human experiments. the main feature of NiceWebRL is that it pre-computes every possible state on the server-side given a initial observation/state. The authors showcase 3 use cases for NiceWebRL (i) Human-Like RL on maze girdworld and Craftax, (ii) Human-AI Collaboration on Overcooked and (iii) Human-Assistive AI on XLand-Minigrid.

**Strengths:**

- The authors provide compelling use cases of how NiceWebRL would be convinient for reserachers in deploying human-AI experiments. I believe this tool would be valuable to share with the ICLR community.
- The paper is well written and easy to follow.

**Weaknesses:**

- The fact that all possible states have to pre-computed on the server-side inherently limits NiceWebRL to only environments with discrete action spaces.
- NiceWebRL only supports environments that already has a Jax implementation, though this will be less of an issue as more and more RL environments become Jax-compatible.

**Questions:**

- How would NiceWebRL support environments with continuous action spaces?

---

### Note · Authors · 2025-12-04

I have read and agree with the venue's withdrawal policy on behalf of myself and my co-authors.